

# Updated trends of the stratospheric ozone vertical distribution in the 60°S-60°N latitude range based on the LOTUS regression model

Sophie Godin-Beekmann[1], Niramson Azouz[1], Viktoria Sofieva[2], Daan Hubert[3], Irina Petropavlovskikh[4], Peter Effertz[4], Gérard Ancellet[1], Doug. A. Degenstein[5], Daniel Zawada[5], Lucien Froidevaux[6], Stacey Frith[7], Jeannette Wild[8], Sean Davis[9], Wolfgang Steinbrecht[10], Thierry Leblanc[11], Richard Querel[12], Kleareti Tourpali[13], Robert Damadeo[14], Eliane Maillard-Barras[15], René Stübi[15], Corinne Vigouroux[3], Carlo Arosio[16], Gerald Nedoluha[17], Ian Boyd[18], Roeland van Malderen[19]

[1]LATMOS Sorbonne Université, UVSQ, CNRS, France,
[2]Finish Meteorological Institute, Finland,
[3]Royal Belgian Institute for Space Aeronomy (BIRA-IASB), Belgium,
[4]Cooperative Institute for Research in Environmental Sciences, U. of Colorado, Boulder, CO, USA,
[5]University of Saskatchewan, Canada,
[6]Jet Propulsion Laboratory, California Institute of Technology, Pasadena, USA,
[7]Science Systems and Applications, Inc &NASA Goddard Space Flight Center, USA,
[8]ESSIC/UMD & NOAA/NCEP/Climate Prediction Center, USA,
[9]NOAA Chemical Sciences Laboratory, USA
[10]Deutsche Wetterdienst, Germany,
[11]Jet Propulsion Laboratory, California Institute of Technology, Wrightwood, USA,
[12]National Institute of Water and Atmospheric Research (NIWA), New Zealand,
[13]Aristotle University of Thessaloniki, Greece,
[14]NASA Langley, USA,
[15]Federal Office of Meteorology and Climatology, MeteoSwiss, Switzerland,
[16]Institute of Environmental Physics, Bremen Universität, Germany,
[17]Remote Sensing Division, Naval Research Laboratory, Washington, DC, USA,
[18]Bryan Scientific Consulting, Charlottesville, VA, USA,
[19]Royal Meteorological Institute, Belgium

*Correspondence to*: sophie.godin-beekmann@latmos.ipsl.fr

**Abstract.**

This study presents an updated evaluation of stratospheric ozone profile trends in the 60°S – 60°N latitude range over the 2000 – 2020 period using an updated version of the Long-term Ozone Trends and Uncertainties in the Stratosphere (LOTUS) regression model that was used to evaluate such trends up to 2016 for the last WMO Ozone Assessment (2018). In addition to the derivation of detailed trends as a function of latitude and vertical coordinates, the regressions are performed with the data sets averaged over broad latitude bands, i.e. 60°S–35°S, 20°S–20°N and 35°N–60°N. The same methodology as in the last Assessment is applied to combine trends in these broad latitude bands in order to compare the results with the previous studies. Longitudinally resolved merged satellite records are also considered in order to provide a better comparison with trends





retrieved from ground-based records, e.g. lidar, ozone sondes, Umkehr, microwave and Fourier Transform Infrared (FTIR) spectrometers at selected stations where long-term time series are available. The study includes a comparison with trends derived from the latest REF-C2 simulations of the Chemistry Climate Model Initiative (CCMI). This work confirms past results showing an ozone increase in the upper stratosphere, which is now significant in the three broad latitude bands. The increase is largest in the northern and southern hemisphere midlatitudes, with ~2.2%/decade at ~2.1 hPa, and ~2.1%/decade at ~3.2 hPa respectively, compared to ~1.6%/decade at ~2.6 hPa in the tropics. New trend signals have emerged from the records, such as a significant decrease of ozone in the tropics around 35 hPa and a non-significant increase of ozone in the southern mid-latitudes at about 20 hPa. Non-significant negative ozone trends are derived in the lowermost stratosphere, with the most pronounced trends in the tropics. While a very good agreement is obtained between trends from merged satellite records and the CCMI REF-C2 simulation in the upper stratosphere, observed negative trends in the lower stratosphere are not reproduced by models at southern and, in particular, at northern midlatitudes, where models report an ozone increase. However, the lower stratospheric trend uncertainties are quite large, for both measured and modelled trends. Finally, 2000-2020 stratospheric ozone trends derived from the ground-based and longitudinally resolved satellite records are in close agreement, especially over the European Alpine and tropical regions.

## 1. Introduction

The recovery of the ozone layer has been under scrutiny since the peak of ozone depleting substances (ODS) was reached in the stratosphere at the turn of the 21$^{st}$ century (e.g. Newman et al., 2007) in response to reduced ODS emissions imposed by the Montreal Protocol signed in 1987. First evidence of the impact of decreasing ODS content on ozone levels was provided in WMO (2014) and references therein for ozone in the upper stratosphere. Since then, an upper stratospheric ozone increase has been confirmed by various studies (e.g. Harris et al., 2015, Steinbrecht et al., 2017, Petropavlovskikh et al., 2019). In parallel, chemistry climate models (CCMs) have attributed half of this increase to decreased ODS concentrations and half to upper stratospheric cooling resulting from increased greenhouse gases (GHGs), which slows gas-phase ozone depleting reactions (e.g., chapter 5 of WMO, 2018). In contrast, an ozone increase in the lower stratosphere has not been detected to date, except for some emerging signs in the Antarctic polar region in spring (de Laat et al., 2015, Solomon et al., 2016, Pazmiño et al., 2018).

The issue of ozone evolution and recovery in the lower stratosphere at extrapolar latitudes was investigated by numerous studies in recent years. Ball et al. (2018) considered several long-term satellite combined records and derived trends using the Dynamical Linear Modelling (DLM) method. They found a decline of ozone in the lower stratosphere over the period 1998-2016. This result was challenged by Chipperfield et al. (2019), who argued that the ozone reduction was influenced by short-term dynamical variability at the end of the studied period. On the other hand, Wargan et al. (2018) used an idealized



atmospheric tracer in a chemistry transport model constrained by MERRA-2 re-analyses in order to assess possible mechanisms responsible for the decrease of lower stratospheric ozone. Their analysis detected enhanced isentropic transport
between the tropical (20°S – 20°N) and extratropical lower stratosphere in the past two decades, which could have influenced the downward trends. This result was confirmed by Ball et al. (2020), who investigated ozone, water vapor and temperature trends from both satellite records and CCM simulations performed in the frame of the REF-B2 CCMI, as well as changes in residual circulation upwelling and mixing efficiency from Japanese 55-year Reanalyses (JRA55) and European Centre for Medium-Range Weather Forecasts (ECMWF) ERA Interim reanalyses. These authors showed that observed ozone trends in
the lower stratosphere can be explained by enhanced mixing between the tropics and extratropics and that these trends are not well reproduced by CCM simulations.

The present study is a follow-up of the ozone profile trend analysis performed within the LOTUS activity of the Stratosphere-Troposphere Processes And their Role in Climate (SPARC) programme (Petropavlovskikh et al., 2019) that contributed to chapter 3 of the last WMO/UNEP Ozone Assessment (WMO, 2018) and is referred to as LOTUS19 hereafter. In order to
achieve a consistent interpretation of stratospheric ozone changes, multiple merged satellite and ground-based data records of ozone vertical distribution were collected to perform unified trend analyses. Previously published multiple linear regression (MLR) models were tested on a common ozone data set to evaluate the sensitivity of derived trends to the use of different models for the regression. This enabled the selection of the open source "LOTUS regression model" (https://arg.usask.ca/docs/LOTUS_regression) that is maintained by the University of Saskatchewan. The trends in the vertical
distribution of stratospheric ozone profiles were assessed over the period 1985 – 2016.  A new approach was established for combining the trend estimates from individual satellite-based records into a single best estimate of ozone profile trends representative of the three broad latitude bands: 35° – 60° at southern (SH) and northern hemisphere (NH) midlatitudes, and 20°N – 20°S in the tropics. Special attention was given to the evaluation of trend significance as a function of altitude. The LOTUS19 trend results were compared to those derived from previous studies (e.g., Harris et al., 2015, Steinbrecht et al.,
2017). LOTUS19 found positive trends in the upper stratosphere in the post-ODS peak period (2000 – 2016) for both satellite and ground-based records. Results from merged satellite records showed statistically significant positive combined trends in the northern hemisphere midlatitudes and in the tropics of 2–3% per decade in the ~5–1 hPa pressure range and 1–1.5% per decade in the ~3–1 hPa pressure range respectively. Combined trends were not statistically significant in the upper stratosphere at southern midlatitudes and no significant trends were obtained in the lower stratosphere. The LOTUS19-derived trends in
broad latitude bands were also compared to model trends from CCMI-1 REF-C2 simulations. Both models and merged satellite records showed similar results in terms of trend values and significance, except at southern midlatitudes in the upper stratosphere where model trends were found to be significant.





After LOTUS19, other studies assessed global stratospheric ozone profile trends. Szelag et al. (2020) analyzed the seasonal dependence of stratospheric ozone trends from four merged satellite datasets over the 2000 – 2018 period using a two-step MLR model. They found positive trends in the upper stratosphere at middle and high latitudes, maximizing during the winter, which is consistent with photochemical control of ozone in that region. In the lower and middle stratosphere, negative trends were found in the tropics during all seasons, along with trends of varying sign depending on the season in the northern and southern midlatitudes. Another study by Sofieva et al. (2021) evaluated regional trends from the new merged gridded dataset MEGRIDOP that combines ozone profile data from six limb-viewing satellite instruments. Zonal trend estimates agreed with previously published results. Longitudinally resolved trends showed a zonal asymmetry in the upper stratosphere at high and middle latitudes in the northern hemisphere with larger trends over Scandinavia than over Siberia.

In the present study, we compute trends from updated versions of the merged satellite records used in LOTUS19 and extended to the end of 2020, from the newly available MEGRIDOP merged time series and from updated versions of the ground-based data records selected from the so-called supersite stations of the Network for the Detection of Atmospheric Composition Changes (NDACC), where ozone profile observations are collected using a variety of ground-based techniques. Trends in the ozone vertical distribution over the post ODS peak period are evaluated using an updated version (version 0.8) of the LOTUS regression model.

The paper is organized as follows. Section 2 summarizes the satellite and ground-based records used in the study, while section 3 describes the updated version of the LOTUS regression model that was employed to retrieve trends from the various records. Section 4 displays the different trend results for the merged satellite data sets both as a function of latitude and vertical levels, and combined in broad latitude bands. Comparison between combined trends in broad latitude bands with corresponding LOTUS19 results and with trends computed from the updated CCMI simulations are presented. In addition, trends from ground-based records at selected NDACC supersites are compared to those from longitudinally resolved satellite data. Section 5 discusses improvements in trend retrievals with respect to LOTUS19, while conclusions of the study regarding long-term ozone profile changes are given in section 6.

## 2. Data

This section provides a brief description of the long-term ozone records used for trend retrieval. Readers can refer to Petropavlovskikh et al. (2019) for a more substantial description of the various observational datasets (Chapter 1 of the report).

### 2.1 Merged satellite records

Seven merged satellite records that were extended to December 2020 are used for this study (see also Table 2.2 of LOTUS19). The Global OZone Chemistry And Related trace gas Data records for the Stratosphere (GOZCARDS v.2.20) ozone monthly





mean record includes HALogen Occultation Experiment (HALOE; v19), Aura Microwave Limb Sounder (MLS v4.2), SAGE I (version 5.9), SAGE II (v7) and covers the period 1979 – 2020. HALOE and Aura MLS measurements are adjusted with SAGE II data, which are used as a reference in the overlapping time periods (Froidevaux et al., 2015). Data included in the

Stratospheric Water and OzOne Satellite Homogenized (SWOOSH) merged record are Aura MLS v4.2, UARS MLS v5, UARS HALOE v19, SAGE II v7 and SAGE III v4. The merged product records are homogenised to minimise artificial discontinuities and to account for inter-satellite biases in the record (Davis et al., 2016). The Solar Backscatter Ultraviolet Merged total and profile Ozone Data (SBUV MOD) record includes data from the SBUV predominantly on the NOAA satellite series of instruments and the Ozone Mapping and Profiler Suite – Nadir profiler (OMPS-NP) on the Suomi National Polar-

orbiting Partnership (SNPP). Data from all SBUV instruments except the NOAA-9 instrument and the morning portion of NOAA-14 and NOAA-16 are included, providing a continuous coverage of ozone profiles since 1978. For the merged data set no external calibration adjustments are applied, as in-instrument calibration adjustments have already been applied at the radiance level within the retrieval algorithm. Measurements are averaged during periods when more than one instrument is operational (Frith et al., 2017). Version 8.7 of SBUV MOD is used in this study, which includes refined in-instrument

calibration adjustments (for NOAA-16 through OMPS NP, using NOAA-19 as a reference) and a diurnal correction to account for varying instrument measurement times (Frith et al., 2020; Kramarova et al., 2022). Another approach was adopted for the SBUV Cohesive (SBUV COH) merged dataset that uses much of the same SBUV and OMPS instruments as SBUV MOD, though retains use of Version 8.6 for SBUV processing as was used in LOTUS19. The COH approach identifies a representative satellite for each time period, and examines data for each overlapping period to improve the consistency of some

satellite records with their neighbors. As in LOTUS19, the updated SBUV COH dataset used in this study adjusts NOAA-16, NOAA-17, NOAA-19 to the NOAA-18 SBUV record, and also extends the record to 2020 with OMPS-NP from SNPP (NOAA v2r3 from NOAA/NESDIS) also adjusted to NOAA-18. Early data are minimally adjusted. Nimbus-7 and NOAA-11 are not adjusted, and NOAA-9 used for a minimal time period to fill a data gap is adjusted to NOAA-11 (Wild et al., 2019). SBUV COH uses Version 8.6 for SBUV data and the NOAA v2r3 version of the OMPS-NP retrieval. The merged SAGE-CCI-OMPS

dataset was developed in the framework of the European Space Agency Climate Change Initiative for Ozone (Ozone-CCI). It includes data from SAGE II, several ozone measuring instruments on board the Environmental Satellite (EnviSat), OSIRIS on Odin, ACE-FTS on the SCIence SATellite (SCISAT), and the OMPS - Limb Profiler (OMPS-LP) (Sofieva et al., 2017). The merging method consists in merging long-term deseasonalized anomalies from the individual satellite ozone records. A similar methodology has been used for the merged SAGE-OSIRIS-OMPS time series that includes data from SAGE II, OSIRIS and

OMPS-LP Usask 2D records (Bourassa et al., 2018; Zawada et al., 2018). Finally, the SAGE-SCIAMACHY-OMPS record includes data from SAGE II, the SCanning Imaging Absorption spectroMeter for Atmospheric CHartographY (SCIAMACHY) and OMPS-LP retrieved with U. Bremen code. The merging of SCIAMACHY and OMPS-LP records with SAGE II occultation observations is carried out from zonally averaged monthly anomalies (Arosio et al., 2019).



Compared to LOTUS19, the SAGE-MIPAS-OMPS record that was not extended to 2020 has been replaced by the SAGE-
SCIAMACHY-OMPS record. Most other records were extended to 2020 with no substantial version change. SBUV MOD
now uses the Version 8.7 retrieval algorithm for all data set used. SBUV-COH is the same dataset as used in LOTUS19 through
2010. The NOAA-19 component for 2011 to 2013 has been reprocessed since the LOTUS19 report with enhanced calibrations,
but no algorithm change, and OMPS-NPP extends the data to 2014 to 2020. For comparison with ground-based datasets, we
also use the newly available MErged GRIdded Dataset of Ozone Profiles (MEGRIDOP) record (Sofieva et al., 2021). This
dataset with a resolved longitudinal structure, is derived from the merging of data six limb and occultation satellite: GOMOS,
SCIAMACHY and MIPAS on Envisat, OSIRIS, OMPS-LP, and Aura MLS using a similar methodology as for SAGE-CCI-
OMPS (Sofieva et al., 2021).

**2.2 Ground-based records**

Several NDACC supersite stations were selected for trend comparison with merged satellite records. The term "supersite" is
linked to the multiple ground-based long-term records available at these stations. The selected NDACC stations are Mauna
Loa (MLO) and Hilo (for ozonesondes) in the tropics, Lauder in the southern hemisphere midlatitudes and various stations
located in the vicinity of the European Alps in the northern hemisphere midlatitudes. The so-called Alpine stations are
Hohenpeissenberg, Arosa, Payerne, Zugspitze, Jungfraujoch and Observatoire de Haute-Provence (OHP).

Ground-based measurement techniques used for the comparison include balloon-borne ozonesondes, lidar (light detection and
ranging), microwave radiometer, FTIR spectrometer, and Umkehr profile retrieval from Dobson sunrise and sunset
measurements.

Ozonesondes are small balloon-borne instruments attached to a standard radiosonde. Based on electrochemical sensing
solution, they measure ozone in situ profile from the ground to about 30 – 35 km altitude with a ~150 m vertical resolution
linked to balloon ascent rate and ozone cell response time.  There are several types of ozonesondes and two of them are used
in this study: Electrochemical concentration cell (ECC) and Brewer-Mast (BM). Each ozonesonde is a unique instrument and
biases have been found in ozonesonde records linked to the preparation method, the type of sonde or the sensing solution used,
or even to the batch of instruments purchased from manufacturers. Since 2004, the WMO/GAW-sponsored Assessment of
Standard Operating Procedures of Ozonesondes (ASOPOS) panel has evaluated and intercompared ozonesonde measurements
in the field or in laboratory chambers. The latest ASOPOS2 report (Smit, Thompson et al., 2021) provides recommendations
on sonde preparation steps and measurement protocols, with the objective to achieve by adoption of these guidelines the 5%
uncertainty level in tropospheric and stratospheric ozone requested by satellite and trends communities. Based on ASOPOS
recommendations, ECC ozonesondes measurements records have been homogenized in multiple stations worldwide and the





ECC ozonesonde data used in this study are from the Harmonization and Evaluation of Ground Based Instruments for Free
Tropospheric Ozone Measurements (HEGIFTOM) prepared set of homogenized ozonesonde records.

Lidar is an active remote sensing technique. For the measurement of the ozone vertical distribution it uses the emission of two
laser wavelengths with different ozone absorption cross-sections according to the so-called Differential Absorption Lidar
(DIAL) technique. Pulsed lasers are used in order to obtain range resolved measurements (e.g. Godin-Beekmann et al., 2003,
Leblanc et al., 2016). This study uses lidar ozone profile records extended to 2020.

Microwave ozone radiometers (MWR) detect emission spectra in the millimetre range produced by thermally excited rotational
ozone transitions (e.g. Maillard-Barras et al., 2019).  The ozone profile retrieval is based on the pressure broadening effect of
the emitted line with the use of a priori profile and optimal estimation method (Rodgers et al., 2000, Bernet et al., 2019).

Umkehr ozone profiles are retrieved from the difference in zenith sky intensities selected from two spectral regions in the so-
called C-pair at 311.5 and 332.4 nm of Dobson and Brewer spectrometer measurements. The long-term record of Umkehr
measurements from four NOAA Dobson spectrophotometers located in the Boulder, OHP, MLO and Lauder stations has
recently been reprocessed with the optimized homogenization technique (Petropavlovskikh et al., 2022) and the three latter
improved records are used here.

We have also compared trends from ground-based ozone records derived with the LOTUS model with those from FTIR
measurements using another trend model (e.g. Vigouroux et al. 2015). The FTIR ozone measurements are performed over the
600–4500cm$^{-1}$ spectral range, with high-resolution spectrometers, using the sun as source of light under clear sky conditions.
On top of total ozone columns, low vertical resolution ozone profiles can be derived from the temperature and pressure
dependence of the line shapes (Hase et al., 1999). Table 1 summarizes the ground-based measurements performed in the
selected NDACC stations and the length of the record.

**2.3 CCMI Model data**

We have used data from the chemistry–climate models (CCMs) participating in phase 1 of the Chemistry–Climate Model
Initiative (CCMI; Eyring et al., 2013), which are able to capture the coupling between stratosphere and troposphere in terms
of composition and physical climate processes more consistently than previous model generations. We used the REF-C2
simulation, that is a seamless simulation running from 1960 to 2100, with our trend analysis covering the period 1979-2020.
REF-C2 experiments follow the WMO (2011) A1 scenario for ozone depleting substances and the RCP 6.0 for other
greenhouse gases, tropospheric ozone precursors, and aerosol and aerosol precursor emissions. Ocean conditions are either
modeled (from a separate climate model simulation), or internally generated, in the case of ocean-coupled models. The
simulation includes state-of-knowledge historic forcings, with recommendations that the 11-year solar cycle and QBO forcings





were either internally model-generated or nudged from the data set provided by Free University of Berlin. No volcanic forcing
was used in this Reference simulation. For a detailed description of all forcings used in the reference simulations, see Eyring
et al. (2013), Hegglin et al. (2016), and Morgenstern et al. (2017). For the CCMI trend analysis, all necessary proxies were
calculated directly from the pertaining individual model simulations. We calculated the appropriate QBO and ENSO proxies
from the model data (zonal winds and SSTs), and used the external forcings (e.g. 11-year solar cycle) as provided to the
modelling groups taking into account their implementation.

**3. The LOTUS regression model**

An updated version of the LOTUS regression model (version 0.8.0) is used for the trend computation. It relies on the classical
multiple linear regression method, which estimates the variability of time series from explanatory variables from the general
least squares approach. The explanatory variables or proxies used in the LOTUS model are the quasi-biennal oscillation
(QBO), the El Niño-Southern Oscillation (ENSO), the 11-year solar cycle, the aerosol optical depth (AOD) and the long-term
trend. As in LOTUS19, we use independent linear trend (ILT) terms to evaluate long-term changes before and after the ODS
peak, e.g., before January 1997 and after January 2000. The LOTUS model is applied to the ozone records without weight
based on e.g. measurement uncertainty. For the datasets that are not deseasonalized, the model includes Fourier components
representing annual and semi-annual variations. The model is based on the following equation for deseasonalized time series:

$$y(t) = A.QBO_1(t) + B.QBO_2(t) + C.ENSO(t) + D.AOD(t) + E.Solar(t) + F.Linear_{pre}(t) + G.Linear_{post}(t) +$$
$$H.C_1(t) + I.C_2(t) + J.C_3(t) + \varepsilon(t) \tag{1}$$

In equation (1), the coefficients A – J are the result of the regression and $QBO_1$ and $QBO_2$ represent two orthogonal components
of the QBO from principal component analysis. No lag is applied to the ENSO and Solar F10.7 proxies, which are taken from
http://www.esrl.noaa.gov/psd/enso/mei/table.html and http://www.spaceweather.ca/data-donnee/sol_flux/sx-5-mavg-eng.php
respectively. For the trend terms, linear components (e.g. $Linear_{pre}$ and $Linear_{post}$) and constant (i.e., $C_1$ and $C_2$) terms are
applied for the "pre" (i.e., before Jan 1997) and "post" (Jan 2000 – Dec 2000) time periods, with zero values outside these
periods, which makes it possible for the two trends to be treated independently of each other. A constant ($C_3$) term is added to
fill the 3-year gap between the "pre" and "post" time periods. $\varepsilon(t)$ represents the residual term. The Cochrane and Orcut (1949)
method is applied to correct for autocorrelation of residuals. See http://argpages.usask.ca/docs/LOTUS_regression/index.html
and LOTUS19 for more information on the LOTUS model.

Several improvements were made to the LOTUS model used in this work compared to the version used in LOTUS19. The
new version (v0.8.0) of the model includes seasonal variation of the predictors (e.g. ENSO, Solar cycle and QBO), which





improves the characterization of ozone variability by the model (section 5) and a new AOD predictor from the GloSSAC
climatology instead of the NASA Goddard Institute for Space Science (GISS) AOD. GloSSAC is taken from
https://asdc.larc.nasa.gov/project/GloSSAC.

The improved LOTUS model with seasonal variation of proxies was applied to the merged satellite records included in the
study over the 1985 – 2020 period for all latitude bins and altitude/pressure levels (depending on the native coordinates of the
time series). It was also applied to each vertical level of the ground-based data used for comparison at the selected supersite
and to the gridded satellite data (e.g. MEGRIDOP, SWOOSH and SBUV MOD) in the vicinity of the stations. Trends from
CCMI model data were obtained from the updated LOTUS model in the same way.

## 4. Results

### 4.1 Global trends as a function of altitude/pressure and latitude from merged satellite records

Figure 1 displays the trend results for the post-2000 period (i.e., from 01/2000 to 12/2020) retrieved from the merged satellite
records for all latitude bands and vertical bins. The upper (bottom) panel shows trend results for records in pressure (altitude)
levels respectively. Dotted areas indicate trend values that are not significant at 2-sigma uncertainty. As in LOTUS19 (e.g.,
Figure 5.2 of the report), positive and significant trend values are observed in the upper stratosphere for all datasets. Some
discrepancies in the magnitude and latitude of the significant positive trends can be noticed among the records. In the upper
panel, the SBUV MOD record show positive trend values around 8 hPa and above 2hPa, while positive trends of the other
records are observed above 7 – 5 hPa and are generally significant for all latitude bands. Both SBUV MOD and SBUV COH
display non-significant trends in the tropical and subtropical latitudes (above 2 hPa in the SBUV COH case). SWOOSH and
GOZCARDS, which share similar individual satellite records, show similar trend patterns, with SWOOSH trend values slightly
larger at 5 hPa at midlatitudes. For records in the bottom panels, the various trends are also very similar in the upper
stratosphere, with increasing trend values from left to right panels. It is also interesting to see significant positive trend values
for most records except SBUV COH in the southern midlatitudes in the middle stratosphere (above ~25 km), while positive
trends are not observed in this region in the northern hemisphere. In the lower stratosphere, trend values are generally negative
but not significant, except in the lowermost stratosphere in the tropics, and especially in the bottom panel. These results are
close to those of LOTUS19. The main feature of the zonally resolved trends is that most combined satellite records now show
significant positive trends in vertical levels between ~ 5 – 2 hPa for all latitude bands. This was not the case in LOTUS19, in
the case of e.g. GOZCARDS, SWOOSH, SAGE-CCI-OMPS for which the trends were not statistically significant in the
tropics.



## 4.2 Trends over broad latitude bands

As in previous works, e.g. Harris et al. (2015), Steinbrecht et al. (2017) and LOTUS19, we investigated trends over broad latitude bands, namely 60°S–35°S, 20°S–20°N, and 35°N–60°N. For GOZCARDS, SWOOSH, SBUV MOD, and SBUV
COH, we first computed the deseasonalized monthly anomalies with respect to their own 1998–2008 climatology for each latitude and pressure bin, then averaged these anomalies over the broad latitude band with latitude weights. The SAGE-SCIA-OMPS, SAGE-CCI-OMPS and SAGE-OSIRIS-OMPS datasets were provided as deseasonalized records with the entire time period of the record used to compute the climatology. Ozone anomalies were averaged in a similar way as in the previous case. The LOTUS model was then applied to each broadband anomaly record. Figure 2 displays the results for the seven merged
records, with each record plotted with respect to its native vertical coordinate and 2-sigma uncertainty. It generally confirms the significant positive trends of ozone for all records in the upper stratosphere, e.g. between ~7 hPa and ~2 hPa in the three broad latitude bands, except for SBUV MOD in the tropics, where the trend is only slightly positive and not significant. Maximum positive trend for this record is seen at around 8 hPa in this region. Notwithstanding SBUV MOD behaviour in the tropics, the spread of trend values is more pronounced in the southern hemisphere where the lowest (largest) trend values are
obtained from SAGE-CCI-OMPS and SAGE-OSIRIS-OMPS respectively. Below 10 hPa, trends are generally close to zero, except in the southern midlatitudes were some positive significant trend values are noticed, e.g. from SAGE-OSIRIS-OMPS, SAGE-SCIAMACHY-OMPS and SWOOSH. In the lowermost stratosphere, e.g. below 20 km, we see a hint of negative trends. This is most pronounced in the tropics but error bars are too large to conclude to a significant decrease. Note also the difference between the northern and southern hemisphere where negative trend values are larger in absolute value in the former
albeit non-significant. Compared to LOTUS19 (e.g. Figure 5.6 of the report), the agreement between the records is much improved, especially in the upper stratosphere due to the lower trend values of SBUV COH, which now agrees quite well with the other records.

## 4.3 Combined trends

The various trend profiles in broad latitude bands were combined in order to facilitate comparison with LOTUS19 results and
with CCMI simulations. For this process, we adopted the same methodology as in LOTUS 19. The combined trend corresponds to the unweighted mean of the seven trend profiles shown in Figure 2. Below the 50 hPa level the mean combines the results from five data records since SBUV data in the lowermost stratosphere are not considered here. The average is done after converting the trends in altitude coordinate to pressure coordinate using a climatological ERA-INTERIM pressure – temperature profile over the period. For the combined trend uncertainty, we have to take into account the correlation between
the individual trend estimates, which is due to the use of common individual satellite records for some of them, e.g. SAGE II, OMPS-LP and the various SBUV time series in the case of SBUV MOD and SBUV COH records. The correlation is also due





to atmospheric variability not characterized by the regression model (see LOTUS19 for more details). The variance of the mean is estimated as follows:

$$\sigma_{mean}^2 = \max\left(\frac{1}{N^2}\sum_{i,j} C_{i,j}\,\sigma_i\sigma_j, \frac{1}{n_{eff}}\sum_i \frac{(x_i-\bar{x})^2}{N-1}\right) \tag{2}$$

where $\sigma_i$ is the uncertainty of individual trends $x_i$ estimated from the fit, $\bar{x}$ is the unweighted mean of the trends, N is the number of averaged records, $C_{i,j}$ are the correlations between the fit residuals for data records *i* and *j*, and $n_{eff}$ is the number of independent values, evaluated as follows:

$$n_{eff} = \frac{N^2}{\sum_{i,j=1}^{N} C_{i,j}} \tag{3}$$

In equation 2, the first term on the right-hand side corresponds to the variance of the mean based on the classical propagation

of errors for correlated variables ($\sigma_{err}^2$) and the second to the variance of the mean for $n_{eff}$ independent estimates ($\sigma_{obs}^2$). The second term considers additional uncertainties in the trend average that are not identified in the first term, such as drifts in the individual time series. More information on this method is given in chapter 5 of LOTUS19. Results of the combined trends from this study, called LOTUS22 hereafter, are displayed in Figure 3 (red curves) with comparison to the LOTUS19 combined trends (blue curve). We show that, compared to LOTUS19, the combined trend uncertainty is significantly smaller, especially

in the upper stratosphere across all three broad latitude bands. This confirms the previous finding that ozone is increasing in the upper stratosphere. The increase is somewhat larger in the NH with a maximum trend of ~2.2%/decade reached at ~2.2 hPa, compared to ~2.1%/decade at ~3.2 hPa in the SH and ~1.6%/decade at ~2.6 hPa in the tropics. Uncertainties of the combined trends are also smaller below 10 hPa, except in the southern hemisphere midlatitudes, where already mentioned positive trends retrieved from some of the records impact both the combined trend average and its uncertainty. It should be noted that slightly

significant negative trends of ~ -1%/decade are retrieved in the tropics in the 30 – 40 hPa range. In the lowermost stratosphere, e.g., below 50 hPa, the LOTUS22 combined trends are negative and systematically larger in magnitude than the LOTUS19 derived trends. In the tropics, the trend uncertainty in our study increases, so that although the negative trends also increase in magnitude, they are not significant, as in LOTUS19. In the SH and NH lower stratosphere, the difference between both combined trends is very small, with in the latter case slightly more negative trends retrieved in this study with also smaller

uncertainty. In all cases and as noted previously, the large uncertainties preclude any definitive conclusion about ozone long-term changes in the lowermost stratosphere.

### 4.4 Trends over super-site NDACC stations

The results of comparisons between ground-based and merged satellite ozone trends for the selected NDACC supersites are

displayed in Figure 4. The merged satellite records are MEGRIDOP, SWOOSH and SBUV MOD, for which longitudinally resolved data were provided. We use the satellite data in the grid cell closest to the stations for the trend computation. For the



ground-based results in the Alpine stations, trends correspond to the average trend of the following records: (1) sondes: Hohenpeissenberg, Payerne and OHP; (2) lidar: Hohenpeissenberg and OHP; (3) Umkehr: Arosa and OHP; (4) FTIR: Zugspitze and Jungfraujoch; (5) MW: Payerne. For MW trends, we used the Payerne record only, as some calibration problems

were found in the MW Bern record. In LOTUS19, ground-based trends were compared to the merged satellite combined trends, which makes direct comparison with the present study more difficult. Figure 4 shows a general good agreement between the ground-based and the gridded satellite trends for the NH Alpine and tropical stations. Trend results are more scattered at Lauder station. Results by the Alpine ground-based instruments reproduce quite well the trend patterns observed with the merged satellite records, e.g. an increase of about 2%/decade on average in the upper stratosphere, trend values around zero in the

middle stratosphere, and mostly negative trends below 20 km, with large uncertainties. It is interesting to see that gridded satellite trend results differ as much as the ground-based ones in the upper stratosphere. At Mauna Loa/Hilo, similar patterns emerge also with an ozone increase in most records in the upper stratosphere, except the MW one, very small negative trends in the middle stratosphere that are most pronounced with the lidar record, and larger negative trends in absolute values in the lower stratosphere below 20 km. For MEGRIDOP and the lidar, negative trends are significant in this altitude region.

Compared to Figure 5.10 of LOTUS 19, sondes and Umkehr results show a better agreement with the other records. Regarding the MW, there were major upgrades to the MLO MW instrument from 2015-2017, including the replacement of the filterbank spectrometer with an FFT spectrometer. At Lauder, a common behaviour of the trend profiles is more difficult to detect due the discrepancies in the results, which were not reduced compared to LOTUS19, despite the homogenization of the sondes and Umkehr records. It should be noted that MW trend results are similar to those displayed in LOTUS19 as the record was not

extended after 2016. Most records show positive trends in the upper stratosphere except the lidar and Umkehr records. Gridded satellite data trends are in between the ground-based ones in this altitude range. In the lowermost stratosphere, negative trends are obtained from most records except from Umkehr. Largest negative trends in absolute values are retrieved with the sondes and lidar records

## 4.5 Comparison with trends derived from CCMI simulations

The comparison of CCMI trend results and merged satellite combined trends (LOTUS22) is displayed in Figure 5, which includes also results from LOTUS19. In the figure, multi-model mean trend estimates from the CCMI-1 REF-C2 simulations are represented by the black line, and the 2-sigma uncertainty of the multi-model mean trend estimates by the grey envelope. Red (blue) solid curves show LOTUS22 (LOTUS19) combined trends, respectively, with corresponding 2-sigma uncertainties represented by the dashed lines. The individual model trends (from a total of 16 CCMI-1 models, as in LOTUS19) are estimated

using the ILT regression method in the same way as for the satellite data and are then combined into a multi-model mean. Model simulations are updated to include 2020, and the necessary proxies are calculated directly from the pertaining individual model simulations where appropriate (e.g., QBO, ENSO), or taken from the external predictors provided to the modelling





groups. Figure 5 shows remarkable agreement between the CCMI and LOTUS22 trend estimates in the upper stratosphere, both regarding the average trend values and the uncertainties. The average agreement is improved compared to the LOTUS19

results in the northern hemisphere. In the middle stratosphere, larger differences are observed, e.g., in the southern hemisphere, where the LOTUS22 trends are positive with large uncertainties, in contrast to CCMI and LOTUS19 trends that are very close to zero. In the lowermost stratosphere, trend values diverge at midlatitudes of both hemispheres, with positive non-significant trends from CCMI. Agreement is better in the tropics, where all results show negative trends although non-significant.

**5. Discussion**

In this section, we discuss in more detail the differences in trend estimates between LOTUS19 and LOTUS22. Compared to Figure 5.6 of the LOTUS19 report, the agreement between the merged data sets is improved, resulting in smaller uncertainties in the combined trend results. This improvement is to a large extent driven by a better agreement of trend results between both SBUV merged records, e.g. SBUV MOD and SBUV COH, and a better agreement with other records' trends, especially in the

upper stratosphere. We can have a better understanding of the improvement in trend uncertainties from Figure 6, which displays the square root of both terms included in the variance of the combined trend (Eq. 2) , i.e. the term linked to error propagation ($\sigma_{err}$, dashed line) and that linked to the standard error of the trend sample ($\sigma_{obs}$, dotted line) for LOTUS22 in red and LOTUS19 in blue. The three panels correspond to the three broad latitude bands. As indicated in Eq. 2, the uncertainty value corresponds to the maximum of both terms as a function of pressure. We can see from the figure that in both studies the uncertainty is

dominated by the $\sigma_{obs}$ term in the upper and middle stratosphere and by the $\sigma_{err}$ term in the lowermost stratosphere. Reduction in the $\sigma_{obs}$ term from LOTUS19 to LOTUS22 is clearly visible in the figure. It is most pronounced in the tropics around 5 hPa, then in the SH in the 5 – 1 hPa pressure range, and in the NH to a somewhat lower extent in the same pressure range. The LOTUS22 $\sigma_{obs}$ term is reduced also in the tropical middle stratosphere but it is increased with respect to LOTUS19 at southern midlatitudes at about the same pressure range. This is due to the already mentioned positive trends retrieved by most of the

records in this latitude and pressure range. In the lowermost stratosphere, the dominance of the $\sigma_{err}$ term was expected due to the large uncertainty retrieved for the trends of the majority of the records in this altitude range.

Another factor that allowed us to reduce the uncertainty of our trend retrieval is the improvement of the LOTUS trend model that includes now seasonal terms for the predictors (Section 3). Thanks to this improvement, the model now better fits the ozone variability of the various records, as shown in Figure 7, displaying adjusted $R^2$ values of the regression in broad latitude

bands for the records with altitude as vertical coordinate on the left and those with pressure on the right. Adjusted $R^2$ provides an estimation of the amount of variance in the monthly data explained by the regression model. It is an indicator of the goodness of the fit. Displayed $R^2$ values correspond to the average of $R^2$ profiles for the 3 latitude bands considered in the study. Solid





(dashed) lines show $R^2$ values retrieved from the LOTUS model with seasonal (non-seasonal) variation of the predictors. Larger $R^2$ values are systematically obtained with seasonal variation of proxies.

Using both the improved merged satellite records for this study and the new version of the LOTUS regression model, we can thus further constrain ozone trends in the various altitude and latitude regions.

## 6. Conclusion

This study provides a new evaluation of stratospheric ozone profile trends in the 60°S – 60°N latitude range from up-to-date
merged satellite and ground-based records. Some satellite data series were improved with respect to those used in the previous assessments (WMO, 2018 and LOTUS19), e.g., SBUV MOD and SBUV COH, which resulted in a better agreement between trends from both records and with the other ones used in the study. Additional records that were absent from the LOTUS19 study are included, e.g., the SAGE-SCIAMACHY-OMPS and MEGRIDOP records. Regarding ground-based data, we use ECC homogenized ozonesonde and Umkehr data reprocessed with optimized homogenization technique that were not
available previously. An updated version of the LOTUS regression trend model allows us to improve the fit of ozone variability for the various records. With these improvements, we can draw the following conclusions:

- The increase of ozone in the upper stratosphere is confirmed, with a clearer recovery in the southern hemisphere compared to LOTUS19. In this altitude region, combined satellite trends are significant in the three broad latitude bands considered, i.e., southern and northern midlatitudes and tropics.
- In the middle stratosphere, e.g., between 50 and 10 hPa, we see the emergence of new signals that will need to be confirmed in the future: an increase of ozone in the southern hemisphere midlatitudes of about 1.2%/decade, though non-significant, and an ozone decrease in the tropics that is (just) significant at around 35 hPa. In the northern hemisphere, ozone trends are close to zero in this altitude range.
- In the lowermost stratosphere, negative ozone trends are obtained for all latitude bands, as in LOTUS19. Trends are
negligible in the southern mid-latitudes. The trends amount to about -2%/decade in the tropics and can still be considered non-significant due to the large uncertainties. Negative ozone trends are also obtained in the northern hemisphere, mainly below 70 hPa. They reach -2%/decade at 100 hPa but are also non-significant.
- Comparison of combined trends with those derived from updated CCMI simulations in broad latitude bands show remarkable agreement in the upper stratosphere, both in trend magnitude and uncertainty. Larger differences are seen
below 10 hPa, 40 hPa and 60 hPa in the tropics, SH and NH midlatitudes, respectively, with the CCMI trends being generally larger than the satellite ones. Differences are most pronounced in the NH midlatitudes, where average satellite trends are negative, while those of CMI are positive but due to the large uncertainty in both cases, these differences are non-significant.

Compared to LOTUS19, we observe a better agreement between trends from ground-based measurements and from satellite records, especially in the tropics and in the Alpine stations of the northern hemisphere. This can be due to the use of longitudinally resolved satellite data and improved ground-based and satellite records. Trends are more scattered at the Lauder station. Examination of the difference between ground-based and merged satellite trends at this location and other NDACC stations should be the subject of a more detailed study in the future.

This study demonstrates that the global ozone observing system is still robust, thanks also to the continuous improvement of
satellite and ground-based records. This allows us to quite accurately evaluate long-term ozone changes in the stratosphere. The cause of larger discrepancies between combined satellite and CCMI trends in the lower stratosphere will have to be further investigated, as the new set of CCMI-2022 set of simulations become available (Plummer et al., 2021). More generally, the study of ozone trends in this region requires a special focus with e.g. new coordinate systems as proposed by the SPARC Observed Composition Trends And Variability in the Upper Troposphere and Lower Stratosphere (OCTAV-UTLS) activity
in order to better constrain ozone variability and provide a more accurate trend evaluation.

**Data availability**

The satellite, ground-based and CCMI monthly mean data are available at the LOTUS FMI depositary: ftp://ftp2.fmi.fi/Phase-2__2022-2019.


**Authors contribution**

NA computed the satellite and ground-based trends, VS computed the combined trends and provided Figure 4, DH provided Figures 3 and 6, KT provided the trends from CCMI simulations, DD, DZ and RD provided and maintain the LOTUS regression model. Other co-authors contributed with satellite or ground-based data. The results of the study were discussed by
all the co-authors. The manuscript was written by SGB with supporting comments by all co-authors.

**Competing interests**

Two co-authors (SGB and IP) are co-organizers of the special issue "Atmospheric ozone and related species in the early 2020s: latest results and trends (ACP/AMT inter-journal SI), 2021".


**Acknowledgements**

The ground-based data used in this publication were obtained as part of the Network for the Detection of Atmospheric Composition Change (NDACC) and are available through the NDACC website www.ndacc.org. Optimized Umkehr data are available at https://gml.noaa.gov/aftp/data/ozwv/Dobson/AC4/Umkehr/Optimized/. Homogenized ozonesonde data were
obtained though the HEGIFTOM Focus Working Group: https://igacproject.org/hegiftom-focus-working-group. NA's work



was supported by a postdoctoral fellowship from Institut National des Sciences de l'Univers du Centre National de la Recherche Scientifique (INSU-CNRS). Work at the Jet Propulsion Laboratory, California Institute of Technology, was performed under contract with the National Aeronautics and Space Administration (80NM0018D004); we gratefully acknowledge the efforts of Ray Wang, John Anderson, and Ryan Fuller towards the initial GOZCARDS ozone data record,

and its subsequent updates. IP and JW were supported by NOAA Climate Program Office's Atmospheric Chemistry, Carbon Cycle, and Climate program, grant number NA19OAR4310169 (CU)/ NA19OAR4310171 (UMD). The SBUV Merged Ozone Data Set was constructed under the NASA MeaSUREs Project and is maintained under NASA WBS 479717 (Long Term Measurement of Ozone). CA acknowledges the support of the University and State of Bremen, and the funding from DAAD and the Living Planet Fellowship SOLVE. The SAGE-CCI-OMPS and MEGRIDOP datasets are created in the framework of

ESA Ozone-CCI + project. The work of VS was supported by European Space Agency (Ozone_cci+ project, contract 4000126562/19/I-NB), the EU Copernicus Climate Change Service for Atmospheric Composition ECVs (contract C3S_312b_Lot2_DLR_2018SC1), and the Academy of Finland (the Centre of Excellence of Inverse Modelling and Imaging (decision 336798).

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



| Station | | Latitude/Longitude | Ground-based records | Record length |
|---|---|---|---|---|
| Alpine | Hohenpeissenberg | 47.8°N/11.0°E | Ozonesonde<br>Lidar | 1966-2020<br>1987-2020 |
| | Payerne | 46.8°N/6.9°E | Ozonesonde<br>Microwave | 1968-2020<br>2000-2020 |
| | Jungfraujoch | 46.5°N/7.9°E | FTIR | 1995-2020 |
| | Arosa | 46.7°N/9.7°E | Umkehr | 1956-2020 |
| | OHP | 43.9°N/5.7°E | Umkehr<br>Lidar<br>Ozonesonde | 1984-2020<br>1985-2020<br>1991-2020 |
| Mauna Loa | | 19.5N°/155.6°W | Umkehr<br>Lidar<br>Microwave | 1984-2020<br>1993-2020<br>1995-2020 |
| Hilo | | 19.7°N/155.1°W | Ozonesonde | 1982-2020 |
| Lauder | | 45°S/169.7°W | Umkehr<br>Lidar<br>Ozonesonde<br>Microwave<br>FTIR | 1987-2020<br>1994-2020<br>1986-2020<br>1992-2016<br>2001-2020 |

**Table 1.** Long-term ground-based NDACC records used in the study.


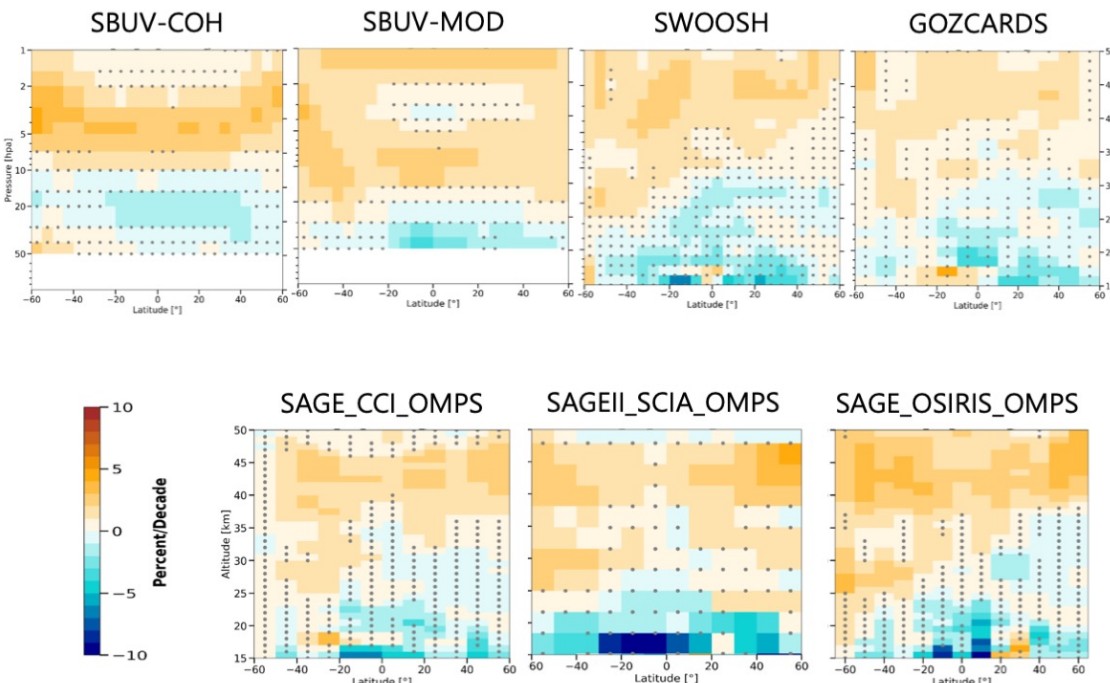

**Figure 1.** Ozone profile trends from merged satellite records in percent per decade for the post-2000 period (Jan 2000 – Dec
2020). Grey stippling denotes results that are not significantly different from zero at the 2-sigma level. Data are presented on
their native latitudinal grid and vertical coordinate




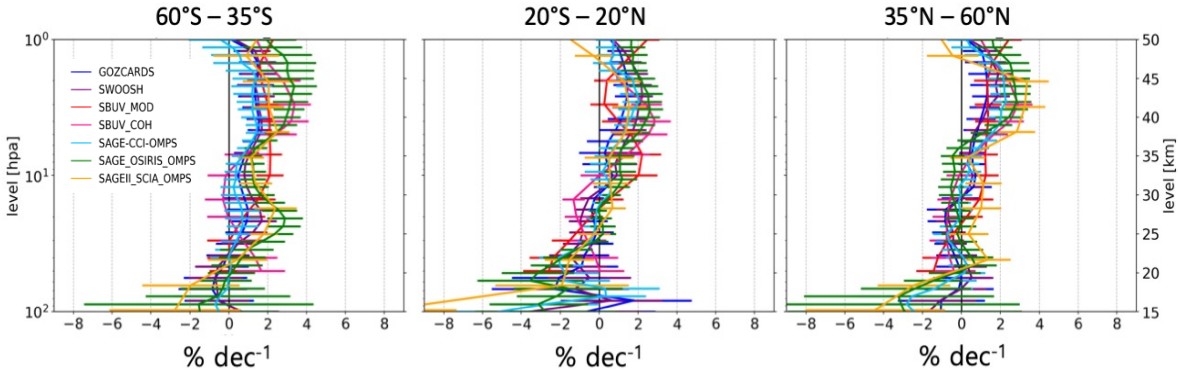

**Figure 2.** Merged satellite ozone trends with their 2-sigma uncertainties for the post-2000 period as estimated by the LOTUS regression model for latitude bands 60°S–35°S (left), 20°S–20°N (center), and 35°N–60°N (right). Colored lines are the trend

estimates from the individual merged data sets on their native vertical grid.


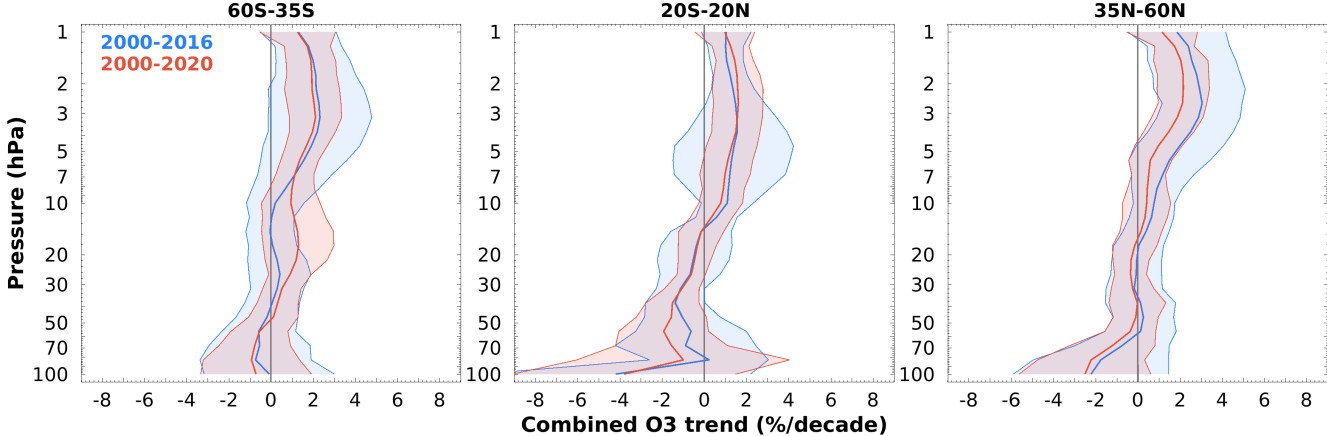

**Figure 3.** Combined post-2000 ozone profile trend estimates and uncertainties (2-sigma) from the seven merged satellite records (below the 50 hPa level: five records, see text). Red (blue) solid line and light red (blue) shaded areas indicate LOTUS22 (LOTUS19) trend values and uncertainties.



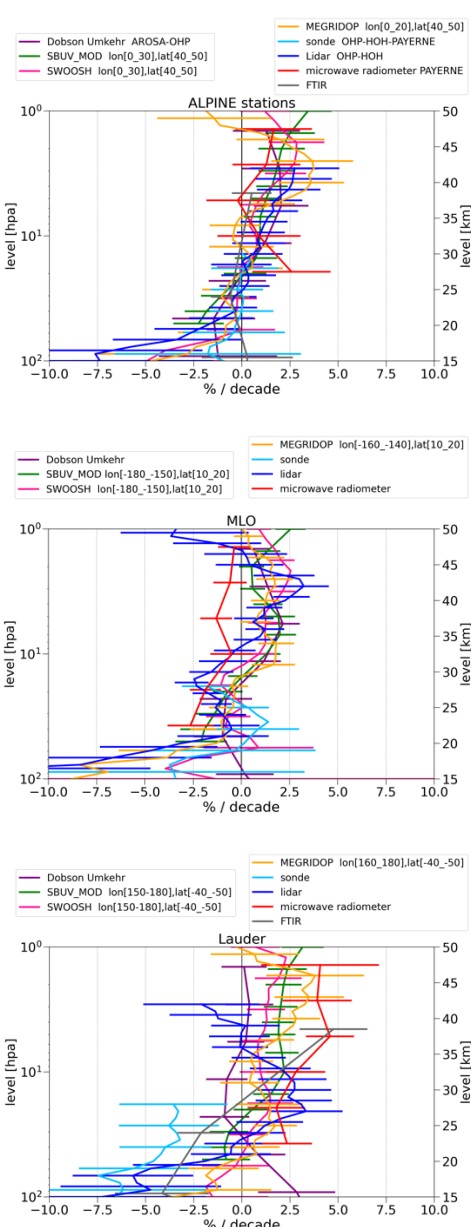

**Figure 4.** Ozone profile trends for the post-2000 periods from ground-based NDACC supersites. Top panel: NH Alpine stations (see text), middle panel: Tropical Mauna Loa and Hilo (ozonesondes) stations, bottom panel: SH Lauder station.



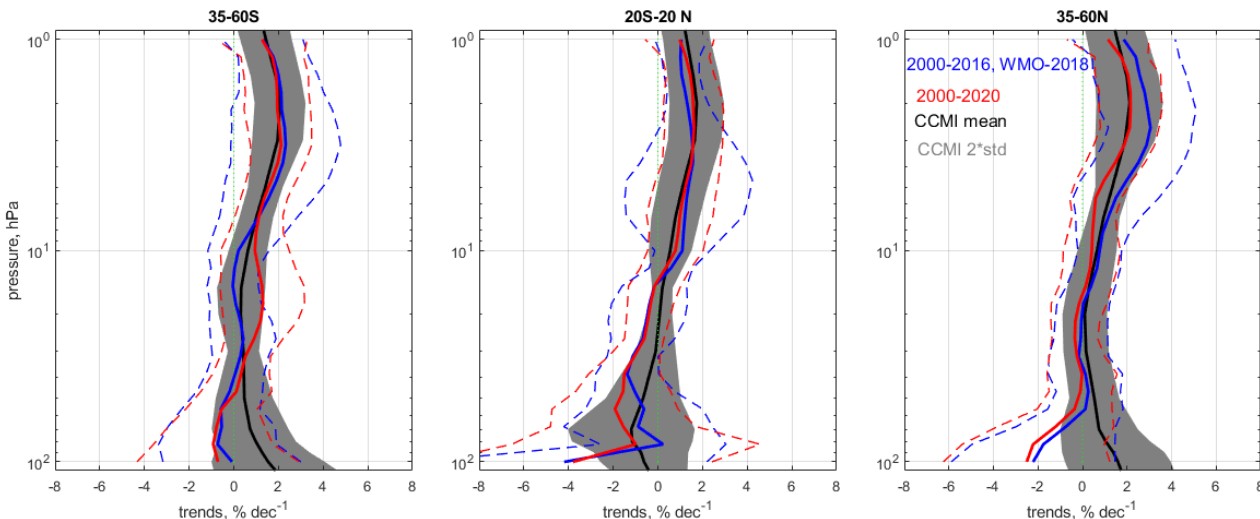


**Figure 5**. Black line: multi-model mean ozone profile trend estimates from the CCMI REF-C2 simulations over three broad latitude bands (left 60°S – 35°S, center 20°S-20°N, right 35°N- 60°N). Grey envelope: 2σ uncertainty of the multi-model mean trend estimates. Red (blue) lines and light red (light blue) dashed lines represent LOTUS22 (LOTUS19) average and 2-
sigma uncertainty respectively.






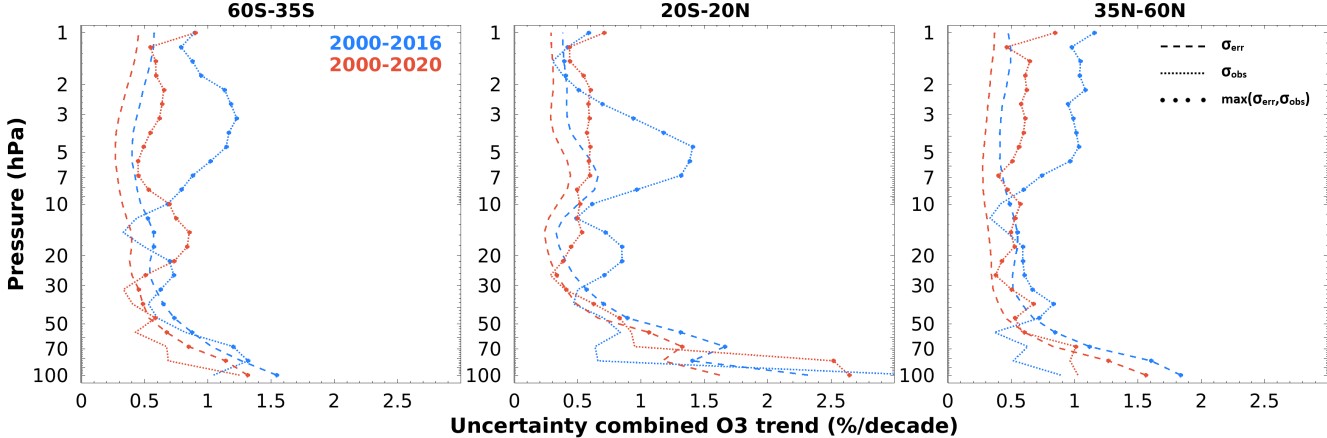

**Figure 6.** Decomposition of error terms for the combine trend estimates: propagation of errors from fit residuals ($\sigma_{err}$, dashed lines) and standard error of the trend sample ($\sigma_{obs}$, dotted lines) for LOTUS22 in red and LOTUS19 in blue for the three broad latitude bands. Combined trend uncertainty is shown by circle symbols that indicate the maximum of $\sigma_{err}$ and $\sigma_{obs}$ in each case.



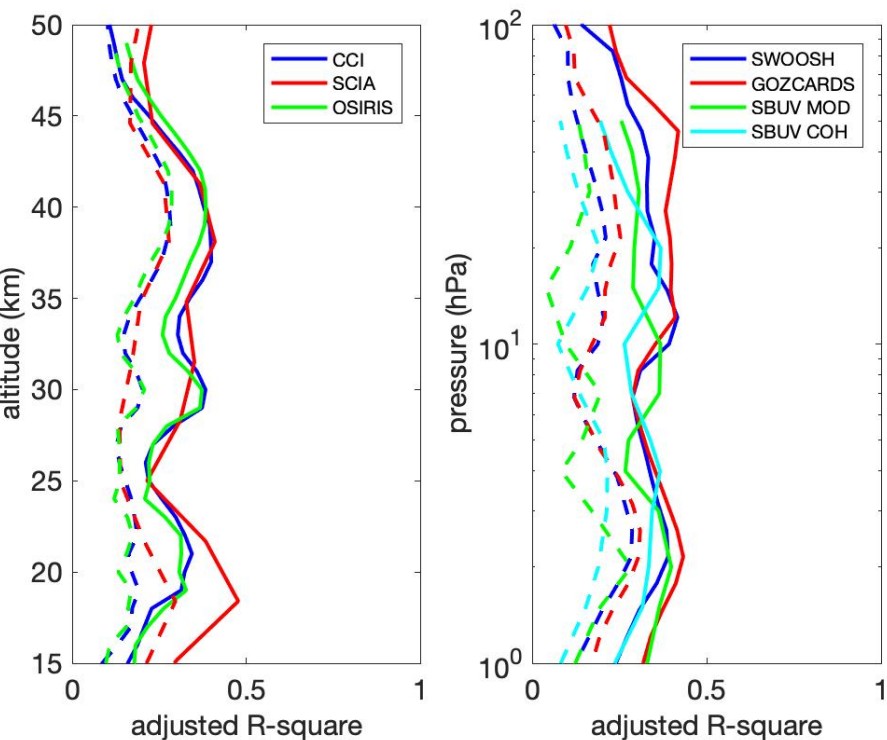


**Figure 7.** Adjusted $R^2$ values of the LOTUS regression model using seasonal (solid lines) and non-seasonal (dashed lines) variation of predictors in the LOTUS regression model (see text).