# Peer review of "Updated trends of the stratospheric ozone vertical distribution in the 60°S-60°N latitude range based on the LOTUS regression model"

_Atmospheric Chemistry and Physics, 2022_

## Referee Comment (RC1)

Review of the study entitled "Updated trends of the stratospheric ozone vertical distribution in the 60°S-60°N latitude range based on the LOTUS regression model" by Godin-Beekmann et al.

The study updates results from the last LOTUS report on ozone trends in the stratosphere and will make an important contribution to the next Ozone assessment report. The text is quite polished and to my view it is ready to be published accepted after some improvements to the figures for better visualization. Well done to the authors. You can find my comments below.

Regarding seasonal variation of proxies: It is concluded that this study gets better results compared to LOTUS19 due to the continuous improvement of satellite and ground-based records and due to the updated version of the LOTUS regression trend model which now takes into account the seasonal variation of the predictors. Some readers may not understand what you mean by "seasonal variation of proxies". The lines 392-399 point to lines 248-249 (section 3) for the seasonal variation of the predictors, but Section 3 does not explain what you mean by seasonal and non-seasonal variation of the proxies. It would be good to clarify how you let the proxy to have a seasonal and non-seasonal variation. If for example, by "seasonal variation of QBO at 30 hPa from 01/2000 to 12/2020" one should understand that it is the monthly time series of equatorial winds at 30 hPa from January 2000 to December 2020, and that the respective non-seasonal variation of QBO is something different, that should be explained. Potentially a Supplement with some graphs might help. My point is that the readers should know how a proxy with seasonal and non-seasonal variation looks like.

Line 297: Interestingly, the trends from SAGEII-SCIA-OMPS data change to negative above 2 hPa which was also not the case in LOTUS19.

Line 427: correct "CMI" to "CCMI".

Line 437: Remove one "set of".

Figs 2 and 4: It is not easy to follow each line between too many error bars. Potentially the error bars could be lighter, and the trend lines thicker. Also, the minor tick marks in vertical axes of pressure [hPa] are barely seen. The minor tick marks in fig 7 cannot be easily seen as well.

Fig 3: Is there improvement when the Ozone-MOD is not included?

Fig 4: I would put the plots side by side as in other figures, i.e., Lauder (left), MLO (middle), Alpine stations (right).

Fig 7: On the vertical axis on the right plot, the $10^2$ is on top of the axis and the $10^0$ is at the bottom. Shouldn't they be upside down?

References: the status of papers marked as "to be submitted" and "in review" should be updated before publication.

---

## Author Comment (AC1)

**Answer to RC1**

We thank anonymous reviewer #1 for reviewing our manuscript and for the very fruitful comments and suggestions, which allowed us to strengthen our article.

**RC1** : Regarding seasonal variation of proxies: It is concluded that this study gets better results compared to LOTUS19 due to the continuous improvement of satellite and ground-based records and due to the updated version of the LOTUS regression trend model which now takes into account the seasonal variation of the predictors. Some readers may not understand what you mean by "seasonal variation of proxies". The lines 392-399 point to lines 248-249 (section 3) for the seasonal variation of the predictors, but Section 3 does not explain what you mean by seasonal and non-seasonal variation of the proxies. It would be good to clarify how you let the proxy to have a seasonal and non-seasonal variation. If for example, by "seasonal variation of QBO at 30 hPa from 01/2000 to 12/2020" one should understand that it is the monthly time series of equatorial winds at 30 hPa from January 2000 to December 2020, and that the respective non-seasonal variation of QBO is something different, that should be explained. Potentially a Supplement with some graphs might help. My point is that the readers should know how a proxy with seasonal and non-seasonal variation looks like.

*Answer: As emphasized also by RC2, the seasonal variation of regressed coefficients in the LOTUS regression model needs some clarification and reformulation. In the former version of the LOTUS regression model used in LOTUS19 report, it was assumed that there was no seasonal dependence of the coefficients retrieved from the regression. In the new version of the model, seasonal terms were added with varying numbers of Fourier components in order to evaluate the seasonal variation of the derived regression model data. Section 3 was rewritten and new equations were added in order to better explain trend computation with the updated LOTUS trend model.*

**RC1:** Line 297: Interestingly, the trends from SAGEII-SCIA-OMPS data change to negative above 2 hPa which was also not the case in LOTUS19.
*Indeed, the SAGEII-SCIA-OMPS record displays a small negative trend above 47km in the 20°S-20°N and 35°N-60°N latitude ranges. These trends are however not significant.*

**RC1:** Line 427: correct "CMI" to "CCMI".
*Done*

**RC1:** Line 437: Remove one "set of".
*Done*

**RC1:** Figs 2 and 4: It is not easy to follow each line between too many error bars. Potentially the error bars could be lighter, and the trend lines thicker. Also, the minor tick marks in vertical axes of pressure [hPa] are barely seen. The minor tick marks in fig 7 cannot be easily seen as well.
*Figures 2 and 4 have been improved accordingly*

**RC1:** Fig 3: Is there improvement when the Ozone-MOD is not included?
*According to Figue 2, SBUV-MOD shows a discrepancy with the records mainly in the upper stratosphere in the 20°S-20°N latitude range and to a lower extend in the middle stratosphere. Since the uncertainty of overall trends in the upper and middle stratosphere is*

*due to spread of individual trend estimates (see Figure 6), exclusion of exceptional value by SBUV-MOD would reduce the uncertainty*

**RC1:** Fig 4: I would put the plots side by side as in other figures, i.e., Lauder (left), MLO (middle), Alpine stations (right).
*Figure 4 has been redrawn accordingly.*

**RC1:** Fig 7: On the vertical axis on the right plot, the 102 is on top of the axis and the 100 is at the bottom. Shouldn't they be upside down?
*Figure 7 has been redrawn accordingly.*

**RC1:** References: the status of papers marked as "to be submitted" and "in review" should be updated before publication.
*This will be down. Thank you for the reminder.*

---

## Author Comment (AC2)

**Answer to RC2 comments**

We thank anonymous reviewer #2 for reviewing our manuscript and for the very fruitful comments and suggestions, which allowed us to strengthen our article.

**RC2**: The chosen approach can be considered quite minimal. The use of a purely linear trend term rather than physical proxies such as GHG concentrations and chlorine and bromine concentrations in different regions of the stratosphere of course restricts the interpretation of the results. It is also an interesting decision that the regression model does not include a term for any large scale dynamical variability beyond QBO and ENSO. This choice also limits the interpretation of the results.

*In order to compare with the previously published trends in LOTUS19, we decided to keep the same methodology and similar proxies for the regressions. We used however the new features added to the LOTUS regression model after the publication of the LOTUS 19 report, such as the seasonal regression of indicators, which improved the significance of the trend terms as described in section 4 of the manuscript. The selection of independent linear functions for the long-term trend terms is explained in detail in chapter 4 of the LOTUS19 report. The following table (table 4.1 of the report) summarizes the advantage and disadvantages of the different long-term trend proxies, e.g. Piecewise Linear Trend (PWLT), Independent Linear Trend (ILT), Singe Equivalent Effective Stratospheric Chlorine (Single EESC) or Empirical Orthogonal Functions of EESC (Two EESC EOFs)*

| MLR Proxy Term | Allows for Curvature? | Allows for Variable Tornaround Time? | Allows for Mono-tonic Trends? |
|---|---|---|---|
| PWLT | No | No | Yes |
| ILT | No | Yes | Yes |
| Single EECS | Yes | No | No |
| Two EESC EOFs | Yes | Yes | Yes |

**Table 4.1:** *Summary of the pros and cons of the different-long-term-ozone-trend proxies.*

*The section 4.3 of the LOTUS19 report concludes: "While both the ILT and EES EOFs are acceptable, the desire to both investigate only the mean trends (i.e., ignore the potential lack of direct correlation between the actual long-term ozone variability and ODSs) and have a more direct analog to compared with results from the last ozone assessments led us to choose the ILT proxy terms."*

*Since the objective of the present study is to provide updated trends, we followed these conclusions. In a similar way, the use of large-scale dynamical proxies (such as the eddy heat flux) in order to better constrain ozone variability was excluded from this study for consistency with the LOTUS19. While we agree that such proxies can indeed improve the significance of derived trends, they also pose other problems such as the selection of meteorological reanalyses to derive them and corresponding artefacts, i.e., spurious long-term trends in the proxy that can in turn influence the final trend results.*

The manuscript itself does seem somewhat rushed in its preparation to me. The core subject (satellite trends) I felt was of a much higher standard than both of the other main topics, namely the comparison with ground-based instruments, or the comparison with models, which are both treated quite cursorily. There is an overall tendency for the text to simply describe the figures without adding much commentary to help the reader draw any

conclusions. Section 5 was an exception and was very welcome, but only considered the core satellite trends.

*The objective of section 4 is indeed to describe the results and point the reader to their major interesting features. We also compared our results with those of the LOTUS19 study in section 4 in order to emphasize agreements or discrepancies. Conclusion paragraphs are now added to subsections 4.4 and 4.5 in order to better emphasize the main results. Section 5 is mainly devoted to the discussion on reason for improvement in trend significance in LOTUS22 compared to LOTUS19. It focuses on satellite trends, which are the main part of the study, since the LOTUS trend model has been applied to a limited number of NDACC ozone profile records at selected stations.*

An example giving the impression of a rushed writing process is that the details of the seasonal variation of proxies were never properly described – section 3 refers the reader to section 5 and section 5 to section 3.

*In fact, the reference to section 5 in line 249 of section 3 was made in order to point the reader to the advantage of using seasonal variation in the trend model. This reference has been removed and the use of seasonal variation of fitted coefficients in the trend model was clarified in section 3, as also suggested by anonymous reviewer #1.*

Another serious omission in this reviewer's opinion is the lack of a comparison with contemporary analysis of total ozone trends (Weber, M., Arosio, C., Coldewey-Egbers, M., Fioletov, V., Frith, S. M., Wild, J. D., Tourpali, K., Burrows, J. P., and Loyola, D.: Global total ozone recovery trends derived from five merged ozone datasets, Atmos. Chem. Phys. Discuss. [preprint], https://doi.org/10.5194/acp-2021-1058, in review, 2022.)

*The article focuses on stratospheric ozone profile trends. Ranged resolved trends are different from total ozone trends since the latter can be influenced also by tropospheric ozone trends as shown by several studies, e.g. Gaudel et al., 2018 or Ziemke et al., 2019. Nonetheless, we have added a mention to Weber et al., 2022 in the conclusion of the paper as follows.*

*"The ozone recovery signal that is mainly observed in the upper stratosphere has an influence on total ozone trends. A recent study indicates a total ozone recovery of $0.5\pm0.2$ %/decade (~1.5 DU/decade) since 1996 (Weber et al., 2022). However, total column ozone evolution is influenced even more by trends in the lower stratosphere and also by tropospheric ozone trends. The latter are estimated to be of the order of ~1.5 DU/decade (e.g. Gaudel et al., 2018; Ziemke et al., 2019) with larger changes found in the tropical regions. The precise impact of stratospheric and tropospheric partial ozone column trends on total column ozone trends needs thus further evaluation."*

*Gaudel, et al., Tropospheric Ozone Assessment Report: Present-day distribution and trends of tropospheric ozone relevant to climate and global atmospheric chemistry model evaluation, Elementa (Wash., DC), 6, 39, 2711 https://doi.org/10.1525/elementa.291, 2018.*

*Ziemke, J. R., et al., Trends in global tropospheric ozone inferred from a composite record of TOMS/OMI/MLS/OMPS satellite measurements and the MERRA-2 GMI simulation, Atmos. Chem. Phys., 19, 3257–3269, https://doi.org/10.5194/acp-19-3257-2019, 2019.*

**RC2:** I wonder if the pre-1997 trends could be included as a supplement? In the LOTUS report this was very useful for providing context to the post-2000 trends.

*We have added the following figure in a supplement of the article.*

[Figure]

*Figure S1. Combined pre-1997 (top panel) and post-2000 ozone (bottom panel) profile trend estimates and uncertainties (2-sigma) from the seven merged satellite records (below the 50 hPa level: five records, see main text). Red (blue) solid line and light red (blue) shaded areas indicate LOTUS22 (LOTUS19) trend values and uncertainties.*

In the supplement, we also included the correlation coefficients of the fit residuals of the seven merged satellite records considered in the study:

| $C_{ij}$ | SBUV MOD | SBUV COH | GOZCARDS | SWOOSH | SAGE-OSIRIS-OMPS | SAGE-CCI-OMPS | SAGE-SCIA-OMPS |
|---|---|---|---|---|---|---|---|
| SBUV MOD | 1.00 | 0.90 | 0.60 | 0.60 | 0.60 | 0.60 | 0.60 |
| SBUV COH | 0.90 | 1.00 | 0.60 | 0.60 | 0.60 | 0.60 | 0.60 |
| GOZCARDS | 0.60 | 0.60 | 1.00 | 0.95 | 0.60 | 0.65 | 0.60 |

| | | | | | | | |
|---|---|---|---|---|---|---|---|
| SWOOSH | 0.60 | 0.60 | 0.95 | 1.00 | 0.65 | 0.70 | 0.65 |
| SAGE-OSIRIS-OMPS | 0.60 | 0.60 | 0.60 | 0.65 | 1.00 | 0.82 | 0.80 |
| SAGE-CCI-OMPS | 0.60 | 0.60 | 0.65 | 0.70 | 0.82 | 1.00 | 0.80 |
| SAGE-SCIA-OMPS | 0.60 | 0.60 | 0.60 | 0.65 | 0.80 | 0.80 | 1.00 |

**Table S1.** Correlation coefficients between the fit residuals of the seven merged satellite ozone profile records considered in the study.

**RC2:** I strongly dislike the use of the term 'supersite' throughout the manuscript, which is both unjustified and ill-advised. Table 1 states of the five European sites, two have only one instrument providing vertically resolved ozone measurements, two have two instruments and only one site has three. What is the basis then for calling them 'supersites'? The clear implication is also that other NDACC sites are in some way inferior (or at least, not as "super") and less worthy of support. Further, figure 4 shows better agreement between the five different but geographically close European sites than between the single sites of Mauna Loa and Lauder, suggesting combining stations is the better way to go anyway for good results.

*The "supersite" has been removed from the manuscript. The idea was initially to select stations where several instruments are implemented in order to monitor various atmospheric parameters, such as Lauder. It is the initial NDACC strategy to promote the use of several measurement techniques at the same station in order to evaluate potential artefacts in the data series. The idea is not to devaluate NDACC stations where only one instrument is implemented. In the case of the "Alpine stations", since their location correspond to one grid cell of the longitudinally resolved satellite measurements, we used the synergy between the various stations. The presentation of the selected NDACC stations in section 2.2 has been modified as follows:*

*"Several NDACC stations were selected for trend comparison with merged satellite records. These stations provide multiple ground-based long-term ozone records using different techniques as mandated by the NDACC strategy (see also [http://ndaccdemo.org/stations](http://ndaccdemo.org/stations))."*

*And at the end of the subsection:*

*"The selected NDACC stations are Mauna Loa (MLO, lidar, microwave, Umkehr) and Hilo (for ozonesondes) in the tropics, Lauder in the southern hemisphere midlatitudes (lidar, ozonesondes, FTIR, Umkehr) and stations located in the vicinity of the European Alps in the northern hemisphere midlatitudes. These Alpine stations are Hohenpeissenberg (lidar, ozonesondes), Arosa (Umkehr), Payerne (microwave, ozonesondes), Zugspitze (FTIR), Jungfraujoch (FTIR) and Observatoire de Haute-Provence (OHP, lidar, ozonesondes, Umkehr). The location of these stations within a radius of less than 700 km corresponds to one*

*grid cell of the longitudinally resolved satellite records used in this study, i.e. 10°lat x 20°long for MEGRIDOP and 10°lat x 30°long for SBUV-MOD and SWOOSH, which facilitates the satellite – ground-based trend comparison."*

**RC2:** Please check that the terminology of upper/middle/lower/lowermost stratosphere is being used consistently throughout the manuscript to refer to the same height regions each time.

*Indeed. This has been checked. The upper (US), middle (MS), lower (LS) and lowermost stratosphere regions correspond to the following altitude/pressure levels:*
*US: Pressures smaller than ~7 hPa, altitudes above ~34 km,*
*MS: Pressures between 20hPa and 7hPa, altitudes between ~26 km and ~34 km,*
*LS: Pressures between 70hPa and 20 hPa, altitudes between ~18 km and ~26 km,*
*LMS: Pressures above70hPa, altitudes below ~18 km,*

**Specific comments**

**Lines 10** 'Finish' should be 'Finnish'

*Done*

**Lines 49-51** Saying "close agreement, especially over the European Alps" is an exaggeration, the agreement is hardly "close" even over Europe while it's worse at Mauna Loa and much worse at Lauder.

*We have modified the sentence as follows:*
*"Finally, 2000-2020 stratospheric ozone trends derived from the ground-based and longitudinally resolved satellite records are in reasonable agreement over the European Alpine and tropical regions, while at the Lauder station in the southern hemisphere mid-latitudes they show some differences."*

**Line 54** Newman et al. 2007 had ODS peaking in the mid 1990s not the 'turn of the century'.

*The sentence was modified in the following way: "The recovery of the ozone layer has been under scrutiny since the peak of ozone depleting substances (ODS) was reached in the stratosphere in the mid and end of the 1990s depending on the latitude region".*

**Line 56** WMO 2010 reported an increase in the NH upper stratosphere, albeit very cautiously.

*The sentence was reformulated: "After first indications of a small ozone increase from various ground-based and satellite records in the upper stratosphere (WMO, 2010), the evidence of the impact of decreasing ODS content on ozone levels in that altitude region was provided in WMO (2014) and references therein".*

**Line 62** You should add WMO 2018 for discussion of Antarctic ozone trends.

*Done*

**Line 63** "Numerous" to me implies more than just four.

*The adjective "numerous" was replaced by "various".*

**Line 72** "performed in the frame of the REF-B2 CCMI" is jargon.

*The reference to this REF-B2 CCMI exercise was removed.*

**Lines 74-76** The reader might struggle to see how 'enhanced mixing' can cause decreasing trends in both the tropics and extra-tropics, which is what you have said.

*The sentence has been changed. Please see our answer to specific comment on lines 63-76.*

**Lines 63-76** This section discusses recent work on trends in the lowermost stratosphere, but the reader might wonder why you don't also give any background for the other altitude regions of the stratosphere

*The issue of ozone trends in the upper stratosphere was addressed in lines 55-60, now lines 60-66 in the present version.*

**Lines 63-76** For this paragraph, personally I would prefer less detail (in particular relating to Ball et al. 2020) and more discussion of how the different results do or don't fit together. In other words, I like the first half of the paragraph better than the second half.

*Following the reviewer's advice, this section was rewritten as follows.*
*"The issue of ozone evolution and recovery in the lower stratosphere has received a lot of attention in recent years. Using several long-term satellite combined records and derived trends based on the Dynamical Linear Modelling (DLM) method, Ball et al. (2018) found a decline of ozone in the lower stratosphere over the period 1998-2016. This result was challenged by Chipperfield et al. (2019), who argued that the ozone reduction was influenced by short-term dynamical variability at the end of the studied period. The cause for ozone decline or lack of recovery in the lower stratosphere was investigated by several model-based studies. Orbe et al. (2020) suggested that the observed decrease of ozone in the northern hemisphere could be explained by a poleward expansion of tropical upwelling and a reduced downwelling in the northern subtropical region. Other studies (Wargan et al., 2018; Ball et al., 2020) pointed to an enhanced meridional mixing between the tropics and the midlatitudes."*

**Lines 91-93** I think this sentence would be clearer if re-arranged to ' … the northern hemisphere mid-latitudes of 2-3% per decade in the 5-1 hPa range and in the tropics of 1-1.5% per decade in the 3-1 hPa range'

*The sentence has been reformulated accordingly.*

**Line 110** The implication of your wording is that the trend is maximum in winter because of photochemical control – the reader might have trouble following the reasoning.

*The reviewer might refer to line 100 of the manuscript. The sentence was rewritten as follows. This is consistent with findings that due to GHG concentration increases, the Brewer-Dobson, which is most effective in the winter season, should strengthen and accelerate the ozone recovery (e.g. Garcia et al., 2008).*

**Line 119** As I wrote earlier, I dislike the use of 'supersite'.

*This term has been removed. Please see our answer to the major comment related to the use of that term.*

**Lines 125 – 167** This section is fine and we don't want any more detail, but nonetheless the big questions seem skipped – (1) are these datasets stable over decades? (2) How successfully is it possible to merge records from different instruments?

*The stability of the merged satellite records is evaluated at length in the chapter 3 of the LOTUS19 report, based in particular on Bayesian analyses. Figure 3.14 of the report compares time series from the merged satellite records used in this study, at different latitudes and altitude ranges. Major inconsistencies in the records are observed prior to 2000. The differences between both merged records derived from the SBUV measurements are also investigated by Frith et al., 2017. In addition, the stability of individual satellite datasets has been extensively analyzed prior to the merging process via various intercomparisons, and also comparisons with ground-based data (e.g., Hubert et al., 2016, Sofieva et al., 2017, Ball et al., 2017). In the merging process, the data quality control is applied. In some merged datasets, some efforts are applied to minimize possible unknown drift of individual satellite records. The success in merging records from different instruments is thus addressed in the articles describing those records and referenced in our paper. Finally, the fact that the merged satellite trend results are generally consistent points to a fairly acceptable stability of the merged satellite records.*

*Ball, W.T., et al., Reconciling differences in stratospheric ozone composites, Atmos. Chem. Phys.,17, 12269-12302, doi:10.5194/acp-17-12269-2017.*

*Frith, S.M. et al.,Estimating uncertainties in the SBUV Version 8.6 merged profile ozone data set, Atmos. Chem. Phys., 17, 14695–14707, 2017, https://doi.org/10.5194/acp-17-14695-2017*

*Sofieva et al., Merged SAGE II, Ozone_cci and OMPS ozone profile dataset and evaluation of ozone trends in the stratosphere, Atmos. Chem. Phys., 17, 12533–12552, https://doi.org/10.5194/acp-17-12533-2017, 2017*

*Hubert, D., et al., Ground-based assessment of the bias and long-term stability of 14 limb and occultation ozone profile data records, Atmos. Meas. Tech., 9, 2497–2534, https://doi.org/10.5194/amt-9-2497-2016, 2016*

**Lines 170-208** Again, do we expect these instruments to be stable over decades?

*Long-term ground-based measurements can be affected by various sporadic issues such as calibration artefacts, e.g. for measurement based on passive remote sensing techniques (MWR, FTIR and Umkehr), malfunction of the instrument (lidar) or in the case of ozone sondes, inadequate preparation method, changes in the sonde type or the sensing solution, or problems in the batch of instruments purchased from manufacturers. The stability of the ground-based records compared to the satellite ones is also investigated at length in the LOTUS19 report and references therein.*

**Line 204** Why would you use a different trend model? That seems very odd. Couldn't the data have been supplied to you and run through your model for consistency?

*The revised version of the article now includes ozone profile trends from FTIR computed with the LOTUS model. The sentence referring to another trend model has thus been removed.*

**Lines 219** Aerosol is a proxy term in your regression model though so won't this introduce an inconsistency? (Is the aerosol term significant in your regression model?)

*Some models did use volcanic forcing, therefore it was used in the analysis. For the CCMI-1 models that did not include it, the analysis returned a minimal insignificant proxy coefficient. sAOD was used as a proxy as provided for the models forcing. The recommendation for the REFC2 set of simulations was not to include volcanic eruptions, although some models did use it. Moreover, the sAOD effects can go through via different routes, e.g. SSTs, QBO.*

**Lines 232-232** It is surprising to me that you don't deseasonalize all datasets so you can treat them consistently. This approach seems to introduce a small but unnecessary inconsistency between the results for the different datasets.

*The satellite and ground-based data sets were generally provided as ozone monthly means. However, in order to compute the trends in broad latitude bands, we first compute the deseasonalized monthly anomalies with respect to their own climatology for each latitude and pressure bin. We then averaged these anomalies over the broad latitude band with latitude weights, in a similar way as in LOTUS19. The impact of the different deasonalization procedure is expected to be small considering the length of the records.*

**Line 240** This link didn't work for me – it gave security warnings. Looking at the website there only seems to be a mixture of daily and monthly values for different time periods available – how did you treat these?

*Links to the source data have been checked and revised. A new table (Table 2) was added to indicate the source of the proxy data.*

**Lines 241-243** It would be interesting to know how well the piecewise linear approach actually could fit the data, ie did the residuals show any systematic departure from the three straight lines?
*For the LOTUS studies, we adopted a classical MLR approach. The figures below show how well the piecewise linear approach fits the data in 2 cases: [60°S-35°S] GOZCARDS broad latitude band at 3.8 hPa and [20°S-20°N] SAGE-CCI-OMPS broad latitude band at 40 km.*

[Figure]

**Lines 274-276** You don't comment on why they have become significant now? Is it four more years of data, improvements in the regression model, improvements in the data or something else?

*Considering the number of records used in the study, we did not explain for each record why trends become significant with respect to LOTUS19. As emphasized in section 2, most records satellite records used in this study were extended to 2020 with no substantial version change, except for SBUV MOD and SBUV COH. So, the change in trends significance is likely due to both data extension and the used of an updated version of the LOTUS model as explained in section 5.*

**Lines 286** – "eg" should be "ie"

*Corrected accordingly*

**Lines 341** "general good" – I wouldn't say "good" – my expectation is the trends from the different instruments would look more similar than this – if this is not the case than more explanation is needed to explain why not.

*We disagree with the reviewer's opinion regarding the results for the Alpine station and the MLO/Hilo stations. Except for the MLO MWR record, which indeed provides different trend results, all other records show a very similar behavior in the vertical trend profiles. The spread between the trend values is of the order of 2%/decade. This is quite good considering the different measuring techniques, and the fact that ground-based trends rely on averages of 1 to 3 individual records at most, which are more affected by geophysical variability.*

**Line 346** This is an example of the text simply describing the figure without adding much to help the reader draw any conclusion.

*We added "which rely on more individual data points" in the sentence in order to emphasize the good agreement between ground-based and merged satellite trends.*

**Lines 351** I don't see how an upgrade in 2015-2017 is going to effect this??

*An instrumental change can have an effect on an ozone record, which needs to be carefully checked in order to avoid step changes in the data series. In the case of MLO data series, we did not detect such change in the record but there is a gap in the record during this period.*

**Lines 355** Well can you use it then? (Or at least, should you put it on the same plot as the others?)

*The MWR record was removed from the figure and the text was modified accordingly.*

**Lines 358** It is very hard for the reader to come away with a clear impression after this discussion. What does it all really mean?

*The following conclusion was added to this section:*
*"In conclusion, trend comparison between ground-based and longitudinally resolved satellite records provide a similar picture as in the previous section, more specifically (1) positive and significant ozone trends in the upper stratosphere from most of the records, (2) negligible trends in the middle stratosphere and (3) negative trends below 20 km, which are statistically significant in lidar and ozonesondes records."*

**Lines 368** Except stratospheric aerosol, right?

*Some models did use volcanic forcing, therefore it was used in the analysis. For the CCMI-1 models that did not include it, the analysis returned a minimal insignificant proxy coefficient.*

**Lines 369** I worry you are "comparing apples and oranges" here. This uncertainty seems completely different to the LOTUS uncertainty. It would be comprised mainly of unforced variability in the models wouldn't it?

*We used exactly the same methodology as in the LOTUS19 studies. As mentioned in the subsection 4.5.2 of the LOTUS19 report, in order to maintain comparability in the interpretation of the results, the trend analysis on the vertical distribution of ozone from the CCMI-1 simulations were performed similarly as for the observation. For the comparison of broad band trend results, all individual ensemble simulations for each model were averaged,*

*so that only one time series for each model/modelling group is included in the average. For trend comparison in broad latitude bands, the deseasonalized model time series were computed as the equally weighted average over the appropriate latitude bands: 60°S–35°S, 20°S–20°N, 35°N–60°N, and 60°S–60°N. This provided a range of model results that can be compared to the range of merged satellite results.*

**Lines 360-374** This section is also very cursory and it is left to the readers to form their own impressions.

*As for the other subsections of section 4, we generally indicated how the results compare with those LOTUS19, which can guide the readers. We added the following conclusion to this subsection.*
*In conclusion, the LOTUS22 trend results confirm the findings of LOTUS19, and provide observed trends that are consistent in magnitude and uncertainty range with simulated ozone trends from the CCMI initiative.*

**Lines 375** I like this section a lot but it is limited in scope.

*Thank you for this appreciation. It is difficult to make the same analysis for the ground-based measurements because LOTUS19 did not use longitudinally resolved satellite data as in this study. This is why we limited this discussion to the merged satellite records.*

**Lines 380-383** This is really good but I think it would be helpful to the reader to make clearer which uncertainties are contributing to these two terms (eg, disagreements between different satellites, instrument drift, lack of ability of the regression model to capture the variability etc).

*In fact, the uncertainties that contribute to the various terms are explained in section 4.3, with also a reference to the section 5.3.1 of the LOTUS19 report where a detailed explanation of the various terms is provided.*

**Lines 434-440** The text seems to be verging into an advertisement for SPRAC activities.

*The sentence was changed as follows: More generally, the study of ozone trends in this region may require a special focus with geophysically-based coordinate systems, based on e.g. tropopause level or equivalent latitude (Millan et al., 2021) in order to better constrain ozone variability and provide a more accurate trend evaluation.*

**Line 615** (Figure 2) Could there be some extra tick marks on the left-hand y-axis (pressure) please?

*Figure 2 has been redrawn.*